# Cisgenderism and transphobia in sexual health care and associations with testing for HIV and other sexually transmitted infections: Findings from the Australian Trans & Gender Diverse Sexual Health Survey

**Shoshana Rosenberg**[1,2], **Denton Callander**[3,4], **Martin Holt**[5], **Liz Duck-Chong**[6], **Mish Pony**[7], **Vincent Cornelisse**[4,8], **Amir Baradaran**[9], **Dustin T. Duncan**[3], **Teddy Cook**[4,6]*

1 Australian Research Centre in Sex, Health and Society, La Trobe University, Melbourne, Victoria, Australia, 2 Centre for Human Rights Education, Curtin University, Perth, Western Australia, Australia, 3 Mailman School of Public Health, Columbia University, New York, New York, United States of America, 4 Kirby Institute, UNSW Sydney, Sydney, New South Wales, Australia, 5 Centre for Social Research in Health, UNSW Sydney, Sydney, New South Wales, Australia, 6 AIDS Council of New South Wales (ACON), Sydney, New South Wales, Australia, 7 Scarlet Alliance, Australian Sex Workers Association, Sydney, New South Wales, Australia, 8 Kirkton Road Centre, Sydney, New South Wales, Australia, 9 School of the Arts, Columbia University, New York, New York, United States of America

* tcook@acon.org.au

## Abstract

Transgender and gender diverse people have unique risks and needs in the context of sexual health, but little is known about sexual health care for this population. In 2018, a national, online survey of sexual health and well-being was conducted with trans and gender diverse people in Australia (n = 1,613). Data from this survey were analysed to describe uptake of sexual health care and experiences of interpersonal and structural cisgenderism and transphobia. Experiences of cisgenderism and transphobia in sexual health care were assessed using a new, four-item scale of 'gender insensitivity', which produced scores ranging from 0 (highly gender sensitive) to 4 (highly gender insensitive). Logistic and linear regression analyses were conducted to determine if experiences of gender insensitivity in sexual health care were associated with uptake and frequency of HIV/STI testing in the 12 months prior to participation. Trans and gender diverse participants primarily accessed sexual health care from general practice clinics (86.8%), followed by publicly funded sexual health clinics (45.6%), community-based services (22.3%), and general hospitals (14.9%). Experiences of gender insensitivity were common overall (73.2% of participants reported ≥2 negative experiences) but most common in hospitals (M = 2.9, SD = 1.3) and least common in community-based services (M = 1.3, SD = 1.4; p<0.001). When controlling for sociodemographic factors, social networks, general access to health care, and sexual practices, higher levels of gender insensitivity in previous sexual health care encounters were associated with a lower likelihood of recent HIV/STI testing (adjusted prevalence ratio = 0.92, 95% confidence interval [CI]:091,0.96, p<0.001) and less-frequent HIV/STI testing (B = -0.07, 95%CI:-0.10,-

**Data Availability Statement:** Data cannot be shared publicly because of their sensitive nature, the potential for individual identification, and ongoing stigma and discrimination enacted against transgender and gender diverse people. Due to these considerations, approval for this research granted by the human research ethics committee of UNSW Sydney (reference: HC180613) and ACON's population-specific human research ethics panel (reference: 2018/21) was provided under the condition that data would not be shared publicly. Data may be provided at request to researchers who agree to the privacy and security provisions of this approval. Such requests can be sent to humanethics@unsw.edu.au.

**Funding:** The authors received no specific funding for this work.

**Competing interests:** The authors have declared that no competing interests exist.

0.03, p = 0.007). Given the high rates of HIV and other STIs among trans and gender diverse people in Australia and overseas, eliminating cisgenderism and transphobia in sexual health care may help improve access to diagnostic testing to reduce infection rates and support the overall sexual health and well-being of these populations.

## Introduction

Transgender and gender diverse people (henceforth 'trans') often encounter a range of barriers to health care, which can significantly undermine access to and quality of care [1–7]. Studies of trans experiences of health care indicate that clinical services and providers are often uninformed of the needs of, or are directly discriminatory towards, trans patients [8,9]. Despite clear and consistent evidence that trans people have unique and often unmet needs in the context of sexual health [10–12], little research has investigated barriers to this kind of care. This paper presents a detailed investigation of sexual health care access and barriers among trans people in Australia.

To understand the kinds of health care barriers that may be relevant to trans people in the context of sexual health, it is important to first assess barriers to health care generally. Research has found that many trans people must navigate fears of stigma and discrimination when accessing health care of any kind [13], including interpersonal forms of transphobia that include being misgendered and being exposed to stigmatizing and discriminatory comments by clinicians and clinical staff [14–18]. These fears are often amplified for trans people of colour forced to contend with intersectional stigma and discrimination [19,20].

Research has shown that most health care providers are ill-prepared to engage meaningfully with the needs of trans patients [21–23]. One systematic review of 20 studies published between 2008 and 2018 found that health care providers often had insufficient knowledge to provide appropriate care for trans patients or made assumptions about trans patients' bodies and health needs [21]. Several other studies have highlighted that lack of knowledge and assumptions often result in an additional demand of labour placed on trans patients to educate providers so that they can receive appropriate health care [24–31]. As an extension of this point, research has also found that health care providers are sometimes so fixated on a patient's trans experience that they fail to attend to the issues for which care was originally sought, what has been referred to as the "trans broken arm" phenomenon [32,33].

There are also several structural barriers to quality health care faced by trans people. Access to health insurance is a known barrier, well documented in the United States [34] but also relevant in the Australian context, as a dearth of publicly funded options for medical gender affirmation diminish access among those without privately funded insurance coverage [35]. The intake paperwork, registration processes, and electronic medical record systems used by many health services often reproduce rigid categories of binary gender and confusion between patient gender, physical characteristics, and gender presumed at birth [36–38]. Many clinics fail to provide gender neutral bathrooms, which is a significant source of concern for many trans people [39] and part of broader organizational tensions that arise through gender segregation in health settings and an overall failure to spatially configure health spaces in ways that are affirming, confidential, and safe for trans patients [40,41].

This previous research presents clear evidence that trans people often encounter barriers to health care, which can be characterized as forms of cisgenderism and transphobia. Cisgenderism (sometimes referred to as cisnormativity) is a form of stigma that denies, ignores, and marginalizes genders other than those that adhere to a fixed gender binary [14], while

transphobia refers to negative feelings, attitudes, or actions directed towards trans people [42]; like other forms of stigma and discrimination, cisgenderism and transphobia can be enacted and experienced on internal, interpersonal, and structural levels [14,42]. Minority stress is theory has been usefully applied in previous work to conceptualize how cisgenderism and transphobia act as barriers to health care: perceived stigma and discrimination result in a stress response, which trans people in turn seek to minimize by limiting engagement in health care as a potential site of further exposure to stigma and discrimination [43–45].

While cisgenderism and transphobia are relevant to all aspects of health care, their implications in the context of sexual health bear special consideration. Sexual health care is so important for trans populations because, globally, they bear a disproportionate burden of HIV and other sexually transmissible infections (STIs) [12,46–50]. Recent evidence also suggests that HIV and STIs uniquely impact Australia's trans populations. In the only national study yet conducted in Australia, HIV was reported among 3.5% of trans men attending sexual health clinics and 5.7% among trans women, which compared with 1.2% among cisgender patients [51]. That study also found that trans women in Australia were 1.5 times as likely to be diagnosed with a bacterial STI than their cisgender peers. Ensuring access to testing and treatment for HIV and other STIs is, therefore, essential towards reducing the rates of these infections and promoting overall sexual health. Little is known, however, about experiences of sexual health care or barriers to HIV and STI testing–including those rooted in cisgenderism and transphobia–among trans people in Australia.

While sexual health can be understood to encompass a wide range of psychological, social, and infection-related considerations, public policy in Australia has largely focused on the prevention, diagnosis, and treatment of HIV and other STIs [52,53]. Previous research suggests that most people in Australia access sexual health care through GP clinics [54,55], while publicly funded sexual health clinics that provide anonymous HIV and STI testing are accessed primarily by nationally defined 'priority populations' like gay men and sex workers [56,57]. Further, recent years have seen the introduction of community-based in some parts of Australia, which provide peer-led HIV and STI testing for trans people. An evaluation of one such service reported largely positive experiences among trans patients, although issues of limited capacity and delayed access to care were noted [58]. While some studies of HIV and STIs have drawn upon clinical samples of trans patients attending GP and sexual health clinics [51,59,60], little is known overall about where trans people in Australia access sexual health care. Additionally, little is known about trans people's experiences of cisgenderism and transphobia while accessing sexual health care in Australia.

To address several prominent gaps in the literature in order to guide health policy and service delivery, the current study sought to provide an assessment of trans people's experiences with sexual health care in Australia. Drawing upon a large, national sample of trans participants, we aimed to describe where trans people receive sexual health care and to document their experiences of cisgenderism and transphobia. Building on a central premise of minority stress theory, we also tested the hypothesis that previous experiences of cisgenderism and transphobia in sexual health care would be negatively associated with HIV and STI testing among trans people in Australia.

## Methods

### Study design

In 2018, a national sample of trans people in Australia was recruited to take part in a cross-sectional, online survey known as the *Australian Trans & Gender Diverse Sexual Health Survey* (www.tgdsexualhealth.com) [10]. Centred on the World Health Organization's holistic

definition of sexual health as a "state of physical, emotional, mental and social well-being in relation to sexuality" [61], the survey sought to collect data to address the dearth of reliable, detailed information on the sexual health and well-being of trans people in Australia and internationally. Data collected through this survey form the basis of the current study.

**Participants and recruitment.** Study recruitment took place over three weeks in October and November 2018. A multi-faceted recruitment strategy was employed, combining online and offline approaches described in more detail elsewhere [58]. To be eligible, participants had to be aged 16 years or older, live in Australia at the time of the survey, and be trans and/or gender diverse. Potential participants were informed that participation was not contingent on whether they had or were planning to undertake gender affirming medical or surgical processes. Those who completed the survey were entered into a raffle to win one of two 300AUD gift cards.

**Measures.** Participants completed a confidential, online survey that included a diverse range of fixed and open-ended questions in the following domains: sociodemographics, social network composition, sexual and romantic practices and experiences, sexual coercion and violence, general health, and experiences of gender affirming processes. Participants were also asked if they had ever received sexual health care from a general practice, sexual health clinic, hospital, and/or community-based service; descriptions and examples of each were provided. Four newly created dichotomous items (no/yes) assessed lifetime experiences of cisgenderism and transphobia enacted within each of the health setting(s) in which participants reported previously receiving sexual health care. Items included general comfort, bodily assumptions by clinical staff, receipt of relevant care, and capacity for proper identification within health systems, and are detailed in Table 1. The survey instrument was reviewed by representatives from organisations supporting the health and well-being of trans people and amended based on their feedback. Further, pilot testing of the survey instrument was conducted with five trans individuals, which included follow-up interviews to capture information on their perceptions and experiences (e.g., readability, interoperability), the results of which were used for further survey refinement.

**Table 1. Experiences [a] of gender insensitivity when receiving sexual health care among a sample of trans and gender diverse people in Australia (n = 1,618), overall and by health setting.**

| | | Setting n (%) | | | | |
|---|---|---|---|---|---|---|
| | Any/all | Comm | SHC | General practice | Hospital | p-value [b] |
| **Previously received sexual health care** | **1,336** | **300** | **613** | **1,166** | **200** | |
| "The intake form allowed me to properly describe my gender experience/history" [c] | 668 (50.0%) | 200 (66.7%) | 323 (52.7%) | 338 (29.0%) | 33 (16.5%) | <0.001 |
| "I felt comfortable disclosing my gender experience or identity" [c] | 985 (73.7%) | 214 (71.3%) | 429 (70.0%) | 714 (61.2%) | 69 (34.5%) | <0.001 |
| "Clinical staff made assumptions about my body or my sex life" | 694 (52.0%) | 79 (26.3%) | 225 (36.7%) | 557 (47.8%) | 131 (65.5%) | <0.001 |
| "I received sexual health care that was sensitive to my individual needs" [c] | 803 (60.1%) | 185 (61.7%) | 364 (59.4%) | 530 (45.5%) | 58 (29.0%) | <0.001 |
| Mean score (SD) [d] | 1.91 (1.23) | 1.27 (1.35) | 1.55 (1.42) | 2.12 (1.34) | 2.85 (1.26) | <0.001 |

SD = standard deviation; Comm = community-based services; SHC = sexual health clinic

[a.] Proportions represent the total number of participants who reported an experience relative to the total number who reported receiving sexual health care within each setting; 282 participants had no previous sexual health care encounter and are not included her

[b.] Differences between health settings were assessed using Chi-squared analyses

[c.] Item reverse coded to generate the total score

[d.] Higher scores = greater gender insensitivity;e.

**Study variables.**   Two primary outcome (i.e., dependent) variables were defined using self-reported data for the 12 months prior to participation: (i) any HIV/STI test, and (ii) number of HIV/STI tests. These variables assessed any reported test for HIV or other STIs and did not distinguish comprehensiveness of testing in terms of infection or anatomical site. Our primary independent variable was defined using responses to the four items on experiences in sexual health care to create an overall measure of 'gender insensitivity' in sexual health care. Gender insensitivity was defined as sexual health care experiences that reinforced or reproduced cisgenderism and/or transphobia on interpersonal and structural levels. Within each health setting, responses to each item (no previous experience = 0, some previous experience = 1) were summed, accounting for reverse-coded items. This was calculated per health setting and overall (i.e., mean of the setting-specific scores), with possible scores of lifetime experiences of sexual health care ranging from 0 (highly gender sensitive) to 4 (highly gender insensitive).

Our analysis included as covariates several sociodemographic characteristics previously identified as relevant to HIV/STI testing in Australia, namely age, area of residence, Indigenous status, and cultural and linguistic diversity [62–65]. Area of residence and cultural and/or linguistic diversity were categorized using standard methods from the Australian Bureau of Statistics [66,67]. In Australia, 'culturally and linguistically diverse' people are defined as those who primarily speak a language other than English or who were born in a country where English is not the primary language [67]. Given significant differences in HIV/STI testing previously identified between cisgender men and women in Australia [68], gender was also included as a covariate. Participants reported 58 different gender labels, which were re-coded using a previously-developed framework as trans men/men, trans women/women, gender non-binary (presumed male at birth) or gender non-binary (presumed female at birth); for the purposes of this analysis, participants who reported both binary and non-binary gender labels were categorized as non-binary [69].

Social networks composition has been shown in research with other populations to influence HIV/STI testing uptake [70,71]; the proportion of participants' social networks reported to be lesbian, gay, bisexual, transgender, queer or some other sexual/gender minority group was, therefore, included as a covariate. Because we expected overall access to health care to influence HIV/STI testing, this factor was included as a covariate (none vs poor/OK vs good/great) along with health insurance status (none/public/private). Public insurance refers to the Australian 'Medicare' program, which is available without cost to all permanent residents and citizens and provides free or subsidised access to some but not all forms of medical care. Further, sexual practices previously associated with HIV/STI testing among other populations (i.e., number of sexual partners, participation in group sex, engaging in sex work or exchange sex, inconsistent condom use with casual partners for vaginal front hole or anal sex) were also included as covariates [72–74], which were self-reported by participants for the 12 months prior to participation.

**Statistical analyses.**   Descriptive analyses of the study variables were performed. By participant gender, bivariate differences in where sexual health care was accessed were assessed using Chi-squared analyses, while ANOVAs were used to assess differences in gender insensitivity. For our primary analysis, generalized linear regression analyses were conducted to investigate the relationship between gender insensitivity in sexual health care (independent variable) and HIV/STI testing, specifically Poisson regression with robust variance (dependent: any HIV/STI test) and linear regression (dependent: HIV/STI test frequency). These analyses were restricted to sexually active participants (i.e., at least one sexual partner in the 12 months prior to participation) to focus on those recommended for HIV/STI testing, with all previously described sociodemographic and behavioural variables included as covariates.

Because the delivery of public health care in Australia is managed primarily by individual states and territories, these analyses also accounted for clustering at the level of jurisdiction.

**Ethical review and community consultation.** The *Australian Trans & Gender Diverse Sexual Health Survey* was reviewed and approved by the human research ethics committee of UNSW Sydney (HC180613). Oversight was also provided by the human research ethics panel of the community organisation ACON, which provides specific ethical input on research involving trans people and other sexual and gender minority groups. The website hosting the survey included information on the study, including any potential risks of participation, a copy of which could be downloaded and retained. Written informed consent was waived for this study, although prospective participants were informed that they could discontinue participation at any point and that this would indicate a withdrawal of consent. Further, participants were informed that completing the survey was an indication of consent. We obtained approval to recruit participants aged 16 and 17 years without parental consent, as relevant aspects of participation (i.e., minor participants were sufficiently mature to understand consent, study was deemed low risk and as having potential benefits, and the potential risks of involving guardians) aligned with Australia's *National Statement on Ethical Conduct in Human Research*.

Several steps were undertaken to involve trans communities in the survey's design and implementation, and the interpretation of findings. First, the study team was comprised of both trans and cisgender investigators. Second, local, state-based and national organisations that support the health and well-being of trans people in Australia were invited to review and provide input into the study design and survey instrument. Third, a meeting of cisgender, trans community partners, researchers, clinicians and policymakers was convened following data collection to conduct a preliminary interpretation of findings, propose key analyses, and outline a dissemination plan.

## Results

In total, 1,920 people started the survey; 1,613 (84.0%) finished the survey and were included in our final sample, which excluded those identified as duplicate respondents and those who did not provide sufficient details with which to classify their gender (Table 2). Overall, participants ranged in age from 16 to 80 years old (M = 30.7, SD = 11.5), nearly half had an undergraduate or postgraduate degree (48.4%), most were Australian born (84.9%) and the majority lived in a major Australia city (82.0%). Our sample included 70 Aboriginal and/or Torres Strait Islander participants (4.3%) and 219 with culturally and/or linguistically diverse backgrounds (13.6%).

In total, 1,336 (82.8%) of participants reported some previous experience receiving sexual health care, including from a general practitioner (86.8%), specialised sexual health clinic (45.6%), community-based service (22.3%), and/or a hospital setting (14.9%). Non-binary participants presumed female at birth were more likely than others to have accessed sexual health care from a general practitioner (91.0%; p<0.001), while sexual health clinics were most commonly attended by trans men (52.4%; p = 0.006) and community-based services by non-binary participants presumed male at birth (28.9%; p = 0.003). By gender, no differences were observed in accessing hospital-based sexual health care (p = 0.585).

Experiences of cisgenderism and transphobia in sexual health care (i.e., 'gender insensitivity') are presented in Table 1; our newly created measure of gender insensitivity demonstrated high internal consistency (Cronbach's α = 0.81). Overall, scores on our measure of gender insensitivity in sexual health care ranged from 0 to 4 (M = 1.91, SD = 1.23), with scores highest among non-binary participants presumed female at birth (M = 2.28, SD = 1.20) and lowest

**Table 2. Sociodemographic, social network, and sexual practice characteristics among a sample of trans and gender diverse people in Australia (n = 1,613) [a].**

| Characteristic | n (%) |
|---|---|
| Age in years, range (M; SD) | 16–80 (30.72; 11.51) |
| Annual income of <40,000AUD | 1,042 (65.2%) |
| Undergraduate or postgraduate degree | 780 (48.4%) |
| Gender | |
| Man/trans man | 258 (16.0%) |
| Woman/trans women | 288 (17.9%) |
| Non-binary, presumed male at birth | 340 (21.1%) |
| Non-binary, presumed female at birth | 727 (45.0%) |
| Intersex | 35 (2.5%) |
| Area of residence | |
| Major city | 1,323 (82.0%) |
| Other | 290 (18.0%) |
| Aboriginal and/or Torres Strait Islander | 70 (4.3%) |
| Culturally or linguistically diverse | 219 (13.6%) |
| Born in Australia | 1,370 (84.9%) |
| Proportion friends who are LGBTQ+ | |
| None or a few | 393 (24.4%) |
| Around half | 321 (19.9%) |
| Most or all | 899 (55.7%) |
| Overall access to general health care | |
| None or poor access | 164 (10.2%) |
| OK or good access | 1,107 (68.7%) |
| Great access | 341 (21.2%) |
| Health insurance | |
| None | 27 (1.7%) |
| Public (i.e., Medicare) | 856 (53.1%) |
| Private | 730 (45.2%) |
| Recent sexual and risk practices [b] | |
| Sexual partner numbers, range (M; SD) | 0–120 (3.08; 9.16) |
| Sexually active [c] | 1,175 (72.8%) |
| Group sex | 241 (14.9%) |
| Sex work and/or exchange sex | 66 (4.1%) |
| Inconsistent condom use (casual partners) [d] | 443 (27.5%) |

a. Missing data: Income (n = 15), intersex status (n = 228), health care access and health insurance (n = 1)

b. 'Recent' refers to the 12-month period prior to participation

c. Sexually active defined as reporting at least one sexual partner in the 12 months prior to participation

d. Defined as 'never' or 'sometimes' using a condom for vaginal, front hole and/or anal sex.

among trans men (M = 1.53, SD = 1.22; p<0.001). Of note, 186 participants (13.9%) had an overall score of 0, indicating they had predominantly experienced gender sensitive sexual health care. By health setting, the highest levels of gender insensitivity were reported when accessing sexual health care in hospital settings (M = 2.86, SD = 1.26) and the lowest levels were in community-based services (M = 1.27, SD = 1.35; p<0.001). Fig 1 provides an overview of gender insensitivity scores stratified by health setting and participant gender.

Details on HIV and STI testing are presented in Table 3. In total, 1,175 participants (72.8%) reported being sexually active in the year prior to the survey, of whom 588 (50.0%) had

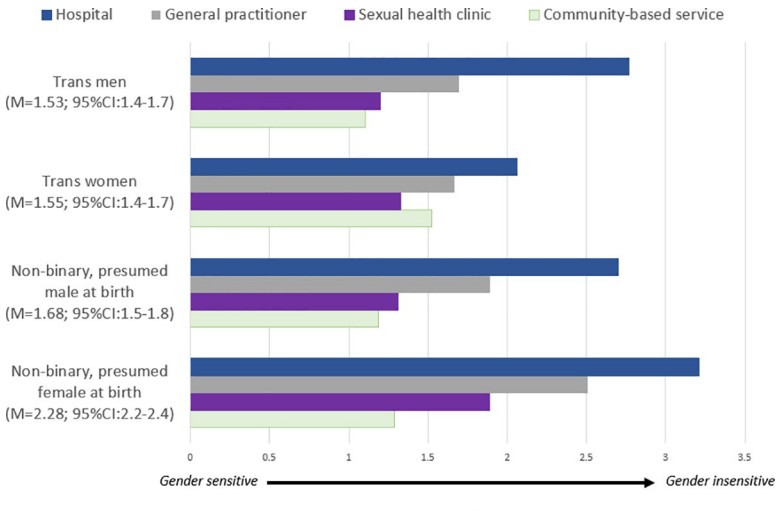

**Fig 1. 'Gender insensitivity' scale scores (mean; 95%CI) comprising experiences of cisgenderism and transphobia when receiving sexual health care among trans and gender diverse participants (n = 1,366) [a], by health setting and gender.** a. Excludes participants who reported no previous sexual health care encounter (n = 252).

received at least one HIV/STI test in the 12 months prior to participation, 339 (28.9%) whose most recent test was conducted more than 12 months prior to participation, and 248 (21.1%) with no previous testing for HIV/STIs. Among participants with a recent HIV/STI test, 302 (28.7%) reported one test in the previous 12 months, 125 (11.9%) two tests, 79 (7.5%) three tests, and 81 (7.7%) reported four or more tests. Among those reporting a test in the previous 12 months, 1.1% self-reported being diagnosed with HIV, 3.1% with chlamydia, 3.2% with gonorrhoea and 0.9% with syphilis.

In the unadjusted analysis, higher levels of gender insensitivity in previous experiences of sexual health care were associated with a lower likelihood of recent HIV/STI testing among sexually-active participants (prevalence ratio [PR] = 0.92, 95% confidence interval [CI]:0.88,0.96, p<0.001), an association that persisted after controlling for covariates (aPR = 0.92, 95%CI:0.88,0.96, p<0.001). Post-hoc power calculations suggest that with the base rate exposure observed for our sample ($\beta 0 = 0.5$) and low correlation between gender insensitivity and the model's covariates ($R^2 = 0.06$), this analysis achieved power exceeding 99% ($\alpha = 0.05$). In the linear regression analysis, higher levels of gender insensitivity were inversely associated with HIV/STI test frequency (B = -0.07, 95%CI:-0.13,-0.00, p = 0.035), which was also the case when controlling for covariates (B = -0.07, 95%CI:-0.10,-0.03, p = 0.007). Post-hoc analyses suggest that with 13 covariates and an observed effect size of 0.35, this analysis achieved >99% Power. Full results from our multivariate analyses are presented in Table 4.

## Discussion

The vast majority of transgender and gender diverse people in our sample reported experiencing cisgenderism and transphobia while accessing sexual health care, and these experiences were associated with a lower likelihood of and less frequent HIV/STI testing. Structural and interpersonal forms of cisgenderism and transphobia were the most commonly reported in hospitals but were reported even in specialized, community-based sexual health settings, a finding that highlights their ubiquity across health settings. Working to reduce and ultimately

**Table 3. HIV and STI testing characteristics among a sample of trans and gender diverse people in Australia (n = 1,613), stratified by sexual activity [a].**

| Characteristic | Recent sexual activity: n (%) | |
|---|---|---|
| | Sexually active [a] (n = 1,175); | Sexually inactive [a] (n = 438) |
| HIV testing history [b,c] | | |
| No previous HIV test | 388 (36.2%) | 251 (63.1%) |
| Recent HIV test | 413 (38.5%) | 43 (10.8%) |
| Non-recent HIV test | 271 (25.3%) | 104 (26.1%) |
| STI testing history [b,d,e] | | |
| No previous ST test | 237 (20.4%) | 236 (55.4%) |
| Recent STI test | 588 (50.5%) | 55 (12.9%) |
| Non-recent STI test | 339 (29.1%) | 135 (31.7%) |
| Number of recent HIV and/or STI tests [b,e] | | |
| No recent test | 457 (43.8%) | 222 (80.4%) |
| 1 test | 302 (28.9%) | 41 (14.9%) |
| 2 tests | 125 (12.0%) | 7 (2.5%) |
| 3 tests | 79 (7.6%) | 3 (1.1%) |
| ≥4 tests | 81 (7.8%) | 3 (1.1%) |
| Recent HIV/STI diagnoses [b,f] | | |
| HIV | 4 (1.0%) | 1 (2.3%) |
| Chlamydia | 19 (3.1%) | 2 (3.4%) |
| Gonorrhea | 22 (3.6%) | – |
| Syphilis | 6 (1.0%) | – |

a. Sexually active defined as reporting at least one sexual partner in the 12 months prior to participation

b. 'Recent' refers to the 12-month period prior to participation

c. Data missing for 143 participants

d. Data missing for 23 participants

e. Any STI test regardless of comprehensiveness by infection or anatomical site)

f. Only among those participants reporting a recent test.

eliminate cisgenderism and transphobia in sexual health care may help increase testing for HIV and other STIs, which has important implications given the disproportionate burden of these infections borne by this population in Australia and internationally [46–51].

Our findings align with those from other research, namely that trans people commonly experience multilevel forms of stigma and discrimination when accessing health care of all kinds [9,75–77] and, per the minority stress model, that these experiences can negatively impact health-seeking behaviours [43–45]. Among our sample, experiences of cisgenderism and transphobia were particularly prominent in 'general' health settings. While this finding may be somewhat unsurprising given the lack of transgender-specific training provided through medical education in Australia [78–81], it is troubling given that general practice clinics were by far the most commonly attended sites for sexual health care in our sample. Although eliminating cisgenderism and transphobia in all health settings is essential, we echo conclusions from other research that general practitioners must be a priority in terms of trans-positive professional development [6,82,83].

It is notable that while cisgenderism and transphobia were less prominent in community-based sexual health services, only around three out of five participants who accessed these services reported receiving care that was sensitive to their needs or could properly describe their gender via intake paperwork. While previous research suggests that this kind of service model

**Table 4. Results of generalized linear regression analyses investigating associations between gender insensitivity [a] and HIV/STI testing among sexually active trans and gender diverse people in Australia (n = 1,175) [b].**

| Variable [reference group, if relevant] | HIV/STI test uptake [c] | | HIV/STI test frequency [d] | |
|---|---|---|---|---|
| | aPR (95%CI) | p-value | B (95%CI) | p-value |
| Gender insensitivity in sexual health care | 0.92 (0.88,0.96) | <0.001 | -0.07 (-0.1,-0.03) | 0.007 |
| Age | 0.99 (0.98,0.99) | <0.001 | -0.01 (-0.02,0.00) | 0.074 |
| Non-binary gender [binary] | 1.01 (0.90,1.13) | 0.849 | -0.08 (-0.27,0.12) | 0.385 |
| Indigenous [non-Indigenous] | 0.89 (0.68,1.17) | 0.871 | -0.12 (-0.43,0.19) | 0.397 |
| Culturally/linguistically diverse [no] | 1.10 (0.94,1.29) | 0.216 | 0.05 (-0.20, 0.31) | 0.655 |
| Live in major city [rural/regional] | 0.99 (0.86,1.15) | 0.905 | 0.04 (-0.29, 0.37) | 0.768 |
| Majority LGBTQ+ social network [minority] | 1.19 (1.02,1.40) | 0.029 | 0.21 (0.12, 0.30) | 0.001 |
| General health care access | 0.88 (0.79,0.98) | 0.016 | -0.19 (0.33, 0.4) | 0.018 |
| Health insurance | | | | |
| None [ref] | – | – | – | – |
| Public | 1.09 (0.57,2.08) | 0.792 | 0.87 (-0.9, 1.8) | 0.070 |
| Private | 1.14 (0.60,2.17) | 0.694 | 0.99 (0.30, 1.96) | 0.045 |
| Recent sexual partner numbers [e] | 1.01 (1.00,1.02) | <0.001 | 0.27 (0.18, 0.37) | <0.001 |
| Recent inconsistent condom use [consistent] [e] | 1.32 (1.19,1.46) | <0.001 | 0.25 (0.84, 0.42) | 0.009 |
| Recent sex work [none] [e] | 1.29 (1.14,1.48) | <0.001 | 0.97 (0.64, 1.31) | <0.001 |
| Recent group sex [none] [e] | 1.47 (1.19,1.63) | <0.001 | 0.68 (0.51, 0.85) | <0.001 |

aPR = adjusted prevalence ratio; CI = confidence interval; RMSE = root mean square error

a. Self-reported experiences of cisgenderism and transphobia, conceptualized as gender insensitivity

b. Complete data for all variables available for 1,044 participants

c. Poisson regression with robust variance, complete data available for 1,045 participants

d. Linear regression, complete data available for 1,044 participants

d. 'Recent' refers to the 12-month period prior to participation.

is viewed favourably and as important by trans people [58], our findings suggest that there is still room for their development and growth towards being truly trans-inclusive. Unfortunately, previous research has also found that demand outstrips capacity for such services and that, in some cases, they are more expensive than attending a general practice [6,21]. Greater funding for community-based sexual and other health services for trans people is, therefore, essential for supporting their expansion and improvement.

We found that gender non-binary participants more commonly experienced cisgenderism and transphobia in sexual health care than binary trans participants. One plausible explanation for this difference is that health systems designed to work best with stability and clear delimitation are more equipped to deal with binary trans patients, especially in contrast to the fluidity and acategorization introduced by non-binary and other forms of gender diversity [69]. Indeed, although in recent years there has been a move to make health settings more inclusive of trans patients, numerous studies have found that these efforts have often not encompassed other gender diverse people [84–86]. There is an important opportunity to build on the momentum thus far focused on including binary trans people to further adapt systems, train staff, and educate clinicians in order to achieve true gender inclusivity. Further research on how to support sexual health care for non-binary and gender diverse people is warranted.

It was observed that having more sexual partners, experiences of condomless sex, and group sex work were all associated with HIV/STI testing uptake and frequency among trans people in our sample. These factors are well-established in clinical guidelines and the literature as increasing the risk of HIV and STIs [87,88], and this finding suggests that our participants

were appropriately assessing their risks and testing accordingly, although it is also possible that their recent testing was in response to a symptomatic infection. It is notable, however, that self-reported diagnoses with HIV and STIs were rare among our sample and much lower than has been reported previously [51], although clinically derived samples often have higher rates of infection than the general public due to symptomatic presentation.

The results of this analysis must also be understood in the context of the study's limitations. First, our sample consisted of participants who were generally well-educated, culturally homogenous, and urban-based. Given the unique and intersectional challenges faced by trans people of colour, those in rural and regional areas, and those of lower socioeconomic status, it seems plausible that the barriers to sexual health care identified through our analysis would be even more prominent among other trans communities. Second, previous research has found that experiences of health care among trans people often differ at the intersections of race and ethnicity [19,20]; unfortunately, an error with how racial and ethnic details were collected from participants prevented their inclusion in this analysis, which is a notable limitation that should be addressed in future research. Third, our measure of gender insensitivity as a proximal marker for cisgenderism and transphobia was newly created and despite extensive community consultation and pilot testing of the survey instrument, its validity and reliability cannot be assumed. Fourth, the cross-sectional nature of our data precludes any assertions of causality, although, as noted, they align with previous theoretical and empirical work related to gender minority stress.

This study represents the first, national investigation of sexual health and well-being among trans people in Australia. Its findings support our hypothesis that experiences of cisgenderism and transphobia can negatively impact HIV/STI testing practices among trans people. As diagnostic testing remains a cornerstone of HIV and STI prevention and management strategies in Australia and globally, attention to reducing cisgenderism and transphobia in sexual health care is essential.

## Author Contributions

**Conceptualization:** Shoshana Rosenberg, Denton Callander, Mish Pony, Amir Baradaran, Teddy Cook.

**Data curation:** Shoshana Rosenberg, Denton Callander, Teddy Cook.

**Formal analysis:** Shoshana Rosenberg, Denton Callander, Amir Baradaran, Teddy Cook.

**Funding acquisition:** Denton Callander, Teddy Cook.

**Investigation:** Shoshana Rosenberg, Denton Callander, Teddy Cook.

**Methodology:** Denton Callander, Teddy Cook.

**Project administration:** Shoshana Rosenberg, Teddy Cook.

**Resources:** Teddy Cook.

**Supervision:** Teddy Cook.

**Visualization:** Denton Callander.

**Writing – original draft:** Shoshana Rosenberg, Denton Callander, Teddy Cook.

**Writing – review & editing:** Shoshana Rosenberg, Denton Callander, Martin Holt, Liz Duck-Chong, Mish Pony, Vincent Cornelisse, Amir Baradaran, Dustin T. Duncan, Teddy Cook.

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
