## [Decision Letter · Decision Letter 0]

12 Feb 2021

PONE-D-20-32098

Cisgenderism, Transphobia, and Their Impact on Australian Trans People’s Experiences of Sexual Health Care Quality and Access: Findings from the Australian Trans & Gender Diverse Sexual Health Survey

PLOS ONE

Dear Dr. Cook,

Thank you for submitting your manuscript to PLOS ONE. After careful consideration, we feel that it has merit but does not fully meet PLOS ONE’s publication criteria as it currently stands. Therefore, we invite you to submit a revised version of the manuscript that addresses the points raised during the review process.

We look forward to receiving your revised manuscript.

Kind regards,

Angelo Brandelli Costa

Academic Editor

PLOS ONE

Journal Requirements:

Reviewers' comments:

Reviewer's Responses to Questions

**Comments to the Author**

1. Is the manuscript technically sound, and do the data support the conclusions?

Reviewer #1: Yes

Reviewer #2: Partly

2. Has the statistical analysis been performed appropriately and rigorously? 

Reviewer #1: Yes

Reviewer #2: No

3. Have the authors made all data underlying the findings in their manuscript fully available?

Reviewer #1: Yes

Reviewer #2: Yes

4. Is the manuscript presented in an intelligible fashion and written in standard English?

Reviewer #1: Yes

Reviewer #2: Yes

5. Review Comments to the Author

Reviewer #1: General comments: I believe that the article placed a great deal of emphasis on issues related to testing for HIV / other STIs. While this is certainly relevant, it does not reflect the entire broad scope of the sexual health research field. I suggest the authors to reframe article's objective, so that it is clear to the reader that the focus is on HIV-related barriers of health care access.

- Introduction:

Please clarify why "most health care providers are ill-prepared to engage meaningfully with the needs of trans patients". Research details should be provided.

Please provide clear definitions on transphobia, cisgenderism and gender insensitivity.

Please provide more details on Australian on sexual health public policy in Australia. Do the clinics only offer HIV/STIs prevention care? Is there any type of care related to reproductive health?

Please provide details on HIV rates on general population and trans population in Australia.

I think gender minority stress model could be described here due to its well documented impact on transgender people's health, especially on HIV/other STIs concerns.

- Methods:

Did the participants signed an informed consent?

The authors mentioned in the introduction the intersectional stigma of trans people of colour. Why did they not include the racial / ethnic status of the participants in the research measures? Including these data in the analyses could add to the literature on racial disparities in health.

Please clarify what "culturally and/or linguistically diverse backgrounds" means.

- Discussion

Data could be discussed taking into account the gender minority stress model, as I mentioned above.

Please provide more insight on why non-binary trans people had more negative outcomes in comparision with binary people.

Final comments: Over all, the article is written in a clear and sound manner, which facilitates reading. As the authors note, trans people often encounter a range of barriers to health care, so this article has the potential to make a useful contribution.

Reviewer #2: I have read with the enjoyment the paper “Cisgenderism, Transphobia, and Their Impact on Australian Trans People’s Experiences of Sexual Health Care Quality and Access: Findings from the Australian Trans & Gender Diverse Sexual Health Survey”. It certainly has many merits, including the scale created and a culturally and linguistically diverse sample. There are some important gaps to be filled in the methods and discussion, which I will describe below. The article would also benefit from better structuring.

The abstract does not include the strongest predictors. Why is it? The introduction clearly states the problem being investigated. However, authors could offer an even greater insight into the topic by considering individual differences. I was wondering if authors have considered the role of other factors, such as personality and particular behaviours that could impede trans people to access proper care. Is there any literature describing these aspects, beyond the well-known role of stigma? Or even the other way round – some behaviours that could enhance health access?

Moreover, the fact that “community-based and peer-led services that provide sexual health care for trans people have been recently introduced in some parts of Australia” could be further explored. I assume there was a good amount of data supporting the opening of these services. I felt that authors could present their hypotheses, as well as could be specific in terms of their study design (as recommended by the Strobe statement).

Methods: Please, include the number of participants and their ages in the “Participants” subheading. The newly created measure needs to be better described (i.e., selection of items, piloting, etc.). Table 1 is rather confusing. If you try to sum the percentages, they extrapolate 100% in columns and rows. Please, explain how the reader must interpret these results. Consider adding Table 1 as supplementary material. In study’s variables, clearly state independent and dependent variables. Place all the covariates under the same subheading, please. Report on missing data, outliers, and how these were handled.

Please, explain why Poisson regression was used in some procedures and not in others. The inclusion of covariates in data analyses seems repetitive since authors mentioned them previously. You did not mention how comparisons displayed in Figure 1 were carried out, as well as assumptions for the test used. Please, report the means with 95 CI in the Figure. Flag in the graph which groups differed. Please, provide fit indices for your regression analyses and the total variance explained. Report on achieved power. Did you compare the proportions in Table 3? Why not? For this purpose, “recent HIV/STI diagnoses” could be grouped together (yes/no). If you do so, update your data analyses section. Why the first predictor in Table 4 is presented in grey background?

The discussion is rather brief and does not explore many interesting aspects from results. For instance, stronger PR were found for sex work and group sex. These same variables were also related to testing frequency. What are the tentative explanations from this, and overall implications for practitioners? The same applies to the other predictors.

6. PLOS authors have the option to publish the peer review history of their article (what does this mean?). If published, this will include your full peer review and any attached files.

Reviewer #1: **Yes: **Ramiro Figueiredo Catelan

Reviewer #2: **Yes: **Guilherme Welter Wendt, PhD

---

## [Author Response · Author response to Decision Letter 0]

1 May 2021

To the editorial board: 

On behalf of my colleagues, please accept our thanks for facilitating peer-review of the above referenced manuscript and for allowing additional time during which to complete our revisions. 

Our thanks extend also to the reviewers, whose comments, suggestions, and questions we have responded to in detail below. Changes to the manuscript have been highlighted in red font, which I believe have resulted in a much more thorough and comprehensive exploration of our study and its findings. 

We are gratified that you are considering this manuscript for publication, and please do not hesitate to be in touch if there are any other ways in which it can be improved. 

Sincerely, 

Denton Callander, PhD

EDITORIAL

Comment i.1: Please ensure that your manuscript meets PLOS ONE's style requirements, including those for file naming.

Response i.1: The manuscript has been reviewed and updated to adhere more closely to PLOS ONE’s style requirements. 

Comment i.2: Please provide additional details regarding participant consent. In the ethics statement in the Methods and online submission information, please ensure that you have specified (1) whether consent was informed and (2) what type you obtained (for instance, written or verbal, and if verbal, how it was documented and witnessed). If your study included minors, state whether you obtained consent from parents or guardians. If the need for consent was waived by the ethics committee, please include this information.

Response i.2: We have provided further details on the processes of informed consent employed by this study and on the inclusion of ‘minor’ participants aged 16 and 17 years. These processes were reviewed and approved by the Human Research Ethics Committee of UNSW Sydney and adhered with Australia’s National Statement on Ethical Conduct in Human Research and are described in the revised manuscript (p 12). 

---

REVIEWER 1

Comment 1.1: I believe that the article placed a great deal of emphasis on issues related to testing for HIV / other STIs. While this is certainly relevant, it does not reflect the entire broad scope of the sexual health research field. I suggest the authors to reframe article's objective, so that it is clear to the reader that the focus is on HIV-related barriers of health care access.

Response 1.1: The reviewer makes an important point. Throughout the manuscript, we have amended our language to more clearly articulate our focus on HIV and STI testing as an important (but by no means all-encompassing) aspect of sexual health care. Notably, we have amended the study title to now read: “Cisgenderism and transphobia in sexual health care and its effects on testing for HIV and other sexually transmitted infections”. 

Comment 1.2:Please clarify why "most health care providers are ill-prepared to engage meaningfully with the needs of trans patients". Research details should be provided.

Response 1.2: Our thanks go to the reviewer for this suggestion. In the revised manuscript, we have included an additional sentence detailing the findings of an influential systematic review, namely that many health care providers have insufficient information to care for trans patient and/or make assumptions about the kind of care they require (p 4). 

Comment 1.3: Please provide clear definitions on transphobia, cisgenderism and gender insensitivity.

Response 1.3: Definitions of cisgenderism and transphobia have been added the manuscript (p 5), as has a definition of gender insensitivity (p 10). 

Comment 1.4: Please provide more details on sexual health public policy in Australia. Do the clinics only offer HIV/STIs prevention care? Is there any type of care related to reproductive health?

Response 1.4: The Introduction section now includes more information on Australia’s public sexual health policies, noting their particular focus on the testing and treatment of HIV and other STIs (p 6). Throughout the manuscript we have also amended our language to more clearly articulate a focus on HIV and STI testing. 

Comment 1.5: Please provide details on HIV rates on general population and trans population in Australia.

Response 1.5: This revised manuscript now includes more information on previous estimates of HIV and STIs among trans people in Australia, which were markedly higher than observed among comparable cisgender populations (p 6). 

Comment 1.6: I think gender minority stress model could be described here due to its well documented impact on transgender people's health, especially on HIV/other STIs concerns.

Response 1.6: The reviewer offers a useful suggestion, for which we are grateful. Information on minority stress and its implications for this project has been added to the revised manuscript (pp 5-6). 

Comment 1.7: Did the participants signed an informed consent?

Response 1.7: As described, we have included additional information on the participant consent processes (p 12). 

Comment 1.8: The authors mentioned in the introduction the intersectional stigma of trans people of colour. Why did they not include the racial / ethnic status of the participants in the research measures? Including these data in the analyses could add to the literature on racial disparities in health.

Response 1.8: We thank the reviewer for this comment and agree that race and ethnicity are important factors in understanding the health of trans people. Unfortunately, an error in how racial and ethnic details were collected from participants in the original survey prevented their meaningful inclusion in this analysis. We have noted this as a potential limitation of this research (p 20). 

Comment 1.9: Please clarify what "culturally and/or linguistically diverse backgrounds" means.

Response 1.9: In Australia, ‘culturally and linguistically diverse’ describes people who primarily speak a language other than English or who were born in a country where English is not the primary language. This information has been added to the revised manuscript (p 10).

Comment 1.10: Data could be discussed taking into account the gender minority stress model, as I mentioned above.

Response 1.10: As described, we have taken gender minority stress into account and included references to it in the revised manuscript. 

Comment 1.11: Please provide more insight on why non-binary trans people had more negative outcomes in comparison with binary people.

Response 1.11: We thank the reviewer for this suggestion and have expanded our discussion on this finding with further detail (p 18). 

---

REVIEWER 2

Comment 2.1: The article would also benefit from better structuring.

Response 2.1: We thank the reviewer for this suggestion and have reorganized several aspects of the manuscript – especially the methods section – for greater overall clarity. 

Comment 2.2: The abstract does not include the strongest predictors. Why is it? 

Response 2.2: The abstract seeks to draw attention to our primary research aim, namely to investigate if experiences of cisgenderism and transphobia were associated with HIV/STI testing. While the reviewer rightly notes that it does not include results relative to other associated factors, these were primarily included in our models as control factors given relationships to HIV/STI testing identified in previous research. We, therefore, feel it is most appropriate to focus the abstract on our primary aim, while providing full details on other associations in the main body. 

Comment 2.3: The introduction clearly states the problem being investigated. However, authors could offer an even greater insight into the topic by considering individual differences. I was wondering if authors have considered the role of other factors, such as personality and particular behaviours that could impede trans people to access proper care. Is there any literature describing these aspects, beyond the well-known role of stigma? Or even the other way round – some behaviours that could enhance health access?

Response 2.3: The reviewer rightly notes that there are many factors that are likely to influence HIV and STI testing, some of which were addressed within our analysis. Given this study’s particular focus on cisgenderism and transphobia, however, we feel that it is most appropriate to focus on the literature that is directly relevant to our primary aim. 

Comment 2.4: Moreover, the fact that “community-based and peer-led services that provide sexual health care for trans people have been recently introduced in some parts of Australia” could be further explored. I assume there was a good amount of data supporting the opening of these services. 

Response 2.4: While we thank the reviewer for this comment, we also note that the only evaluation of community-based sexual health services for trans people in Australia is cited. Unfortunately, empirical research on these services is lacking, a point that further reinforces the impetus for this study. 

Comment 2.5: I felt that authors could present their hypotheses, as well as could be specific in terms of their study design (as recommended by the Strobe statement).

Response 2.5: We thank the reviewer for this suggestion and have more explicitly detailed this study’s hypothesis (p 7). We note that the manuscript includes 8 out of the 9 detail domains suggested by STROBE for cross-sectional research, excluding only ‘Bias’ which was not directly relevant to a convenience sample of this kind (the limitations of which are described in the Discussion section on p 12). 

Comment 2.6: Please, include the number of participants and their ages in the “Participants” subheading.

Response 2.6: While we thank the reviewer for this comment, this manuscript is organized so as present all results – including those relevant to sample size and characteristics – in the Results section. We will defer to editorial guidance as to the best place to situate this information within the overall manuscript. 

Comment 2.7: The newly created measure needs to be better described (i.e., selection of items, piloting, etc.).

Response 2.7: In the revised manuscript we have provided more details on the creation of this study’s survey instrument, including its piloting procedures (p 9). 

Comment 2.8: Table 1 is rather confusing. If you try to sum the percentages, they extrapolate 100% in columns and rows. Please, explain how the reader must interpret these results. Consider adding Table 1 as supplementary material.

Response 2.8: Given the centrality of these items to our overall analysis, we believe it is important that they are presented in the main manuscript as opposed to an appendix. We have, however, added a footnote to Table 1 to help clarify the nature of the presented proportions, i.e., they represent the total number of participants reporting an experience (numerator) relative to the total number who reported receiving sexual health care in a particular setting (denominator). 

Comment 2.9: In study’s variables, clearly state independent and dependent variables. Place all the covariates under the same subheading, please. Report on missing data, outliers, and how these were handled.

Response 2.9: While we have not created new sub-headings for this section, it has been reorganized to present our primary outcome (dependent) variables first, followed by our primary independent variable (i.e., gender insensitivity). Other covariates have been grouped together and presented later in this section (pp 9-10). 

Comment 2.10: Please, explain why Poisson regression was used in some procedures and not in others.

Response 2.10: As outlined in the Methods section, we employed Poisson regression for our dichotomous ‘count’ outcome and linear regression for the continuous ‘frequency’ outcome, as linear regression was deemed most appropriate for conducting a multivariable analysis with an independent variable that was continuous in nature. 

Comment 2.11: You did not mention how comparisons displayed in Figure 1 were carried out, as well as assumptions for the test used.

Response 2.11: Details on the tests used to assess bivariate associations have been added to the revised manuscript (p 11). 

Comment 2.12: Please, report the means with 95 CI in the Figure. Flag in the graph which groups differed.

Response 2.12: We thank the reviewer for this helpful suggestion and have added 95%CIs to the figure, but to avoid oversaturating the graph we have opted not to include the analyses of difference, which are presented instead in the text. 

Comment 2.13: Report on achieved power.

Response 2.13: For both primary analyses, post-hoc calculations suggest that we achieved power exceeding 99%. The details of these calculations have been added to the revised manuscript (pp 16-17). 

Comment 2.14: Did you compare the proportions in Table 3? Why not? For this purpose, “recent HIV/STI diagnoses” could be grouped together (yes/no). If you do so, update your data analyses section.

Response 2.14: Sexually inactive people generally test less than those who are sexually active, and indeed sexual health guidelines typically make no recommendations for HIV or STI testing among people without sexual partners. For these reasons, while we thank the reviewer for this suggestion, we contend that it is superfluous to analyze differences in testing between active and inactive participants. 

Comment 2.15: Why the first predictor in Table 4 is presented in grey background?

Response 2.15: This row was shaded to help differentiate it to readers, but we have removed this colouring to avoid any confusion. 

Comment 2.16: The discussion is rather brief and does not explore many interesting aspects from results. For instance, stronger PR were found for sex work and group sex. These same variables were also related to testing frequency. What are the tentative explanations from this, and overall implications for practitioners? The same applies to the other predictors.

Response 2.16: The reviewer makes an excellent suggestion. Many of the factors identified in our analysis as relevant to HIV and STI testing are commonly understood in sexual health research as predictors of testing, and among our sample they signal that trans people can self-assess their risks and test accordingly. We have expanded the Discussion section to more explicitly attend to these findings (p 19).

---

## [Decision Letter · Decision Letter 1]

9 Jun 2021

Cisgenderism and transphobia in sexual health care and associations with testing for HIV and other sexually transmitted infections: Findings from the Australian Trans & Gender Diverse Sexual Health Survey

PONE-D-20-32098R1

Dear Dr. Cook,

We’re pleased to inform you that your manuscript has been judged scientifically suitable for publication and will be formally accepted for publication once it meets all outstanding technical requirements.

Kind regards,

Angelo Brandelli Costa

Academic Editor

PLOS ONE

Additional Editor Comments (optional):

Reviewers' comments:

Reviewer's Responses to Questions

**Comments to the Author**

1. If the authors have adequately addressed your comments raised in a previous round of review and you feel that this manuscript is now acceptable for publication, you may indicate that here to bypass the “Comments to the Author” section, enter your conflict of interest statement in the “Confidential to Editor” section, and submit your "Accept" recommendation.

Reviewer #1: All comments have been addressed

Reviewer #2: All comments have been addressed

2. Is the manuscript technically sound, and do the data support the conclusions?

Reviewer #1: Yes

Reviewer #2: Yes

3. Has the statistical analysis been performed appropriately and rigorously? 

Reviewer #1: N/A

Reviewer #2: Yes

4. Have the authors made all data underlying the findings in their manuscript fully available?

Reviewer #1: Yes

Reviewer #2: No

5. Is the manuscript presented in an intelligible fashion and written in standard English?

Reviewer #1: Yes

Reviewer #2: Yes

6. Review Comments to the Author

Reviewer #1: Most of my concerns from my previous review have been addressed. I recommend the publication of the revised version of this paper, which will certainly add to the literature.

Reviewer #2: All the issues were properly solved in the revised paper. I have no further suggestions to make, or any other questions regarding the study.

7. PLOS authors have the option to publish the peer review history of their article (what does this mean?). If published, this will include your full peer review and any attached files.

Reviewer #1: **Yes: **Ramiro Figueiredo Catelan

Reviewer #2: **Yes: **Dr. Guilherme Welter Wendt

---

## [Editor Report · Acceptance letter]

28 Jun 2021

PONE-D-20-32098R1 

Cisgenderism and transphobia in sexual health care and associations with testing for HIV and other sexually transmitted infections: Findings from the Australian Trans & Gender Diverse Sexual Health Survey 

Dear Dr. Cook:

I'm pleased to inform you that your manuscript has been deemed suitable for publication in PLOS ONE. Congratulations! Your manuscript is now with our production department. 

Kind regards, 

on behalf of

Dr. Angelo Brandelli Costa 

Section Editor

PLOS ONE